# Svalbard glacier calving front retrievals during the 1960s and 1970s from archived Landsat and declassified intelligence satellite photographs

Loris Danjou<sup>1,2</sup>, Eero Rinne<sup>1</sup>, and Erik Schytt Mannerfelt<sup>1,3</sup>

**Correspondence:** Eero Rinne (eeror@unis.no)

Abstract. We retrieved calving front locations of 171 Svalbard tidewater glaciers from archived satellite imagery. We used early Landsat images from 1976–1978 as well as declassified intelligence satellite photographs from 1962–1963. To support the geophysical analysis of these calving fronts, we also used historical aerial images from 1936–1938. During our study period between 1936 and 1978, we estimate an average glacier retreat rate of  $26.3 \, \mathrm{m} \, \mathrm{yr}^{-1}$ . This corresponds to an approximately 1 km of average retreat of Svalbard tidewater glaciers during this period. By multiplying the retreats by the glaciers' widths, we estimate the cumulative area loss rate to be  $16.0 \, \mathrm{km}^2 \, \mathrm{yr}^{-1}$  ( $R^2 = 0.94$ ), which is slightly lower than the estimates found in the existing literature for the periods 1985-2023 and 2000-2020. Looking at individual glaciers, we identify and discuss 15 significant advance events. For four of them, our study provides additional information to current knowledge. We have discovered one undocumented surge – we show evidence that Emmabreen has surged between 1936 and 1963. We have also narrowed down the time windows of two previously known surges: the surge of Allfarvegen to between 1976 and 1978 (previously reported to have happened between 1970 and 1980), and the surge of Stonebreen to between 1936 and 1963 (previously reported between 1936 and 1971). We also show that Schweigaardbreen and Fonndalsbreen have advanced 400 and 541 meters, respectively, between 1938 and 1976. We discuss the potential future uses of our dataset, consisting of georeferenced satellite images used in our study as well as digitized calving fronts, which is freely available for future research.

#### 5 1 Introduction

#### 1.1 Svalbard glaciers

Svalbard is a Norwegian archipelago located in the high Arctic, between 74° and 81°N latitude, and 10° and 34°E longitude. About 57% of the archipelago is covered by ice; the Randolph Glacier Inventory version 7 (RGI; RGI 7.0 Consortium, 2023) inventoried 1,666 glaciers on the Archipelago, among which 196 terminate in the sea, all of which seem to be grounded (Hagen et al., 1993). Many Svalbard glaciers have a history of surging, meaning that their flow velocities can dramatically increase for a short period of time (typically 1–10 years). The climate, basal rheology, and glacier geometry of typical Svalbard glaciers

<sup>&</sup>lt;sup>1</sup>The University Centre in Svalbard (UNIS), Department of Arctic Geophysics

<sup>&</sup>lt;sup>2</sup>currently at UiT The Arctic University of Norway, Department of Physics and Technology

<sup>&</sup>lt;sup>3</sup>Department of Geosciences, University of Oslo, 0316 Oslo, Norway

45

seem especially favorable for surging, and the archipelago is one of the world's hotspots for surge activity (Sevestre and Benn, 2015). Among Svalbard tidewater glaciers, 139 are of surge-type (Harcourt et al., 2025).

Globally, glaciers are an important component of the climate system. They are a freshwater storage, and their mass loss can dramatically influence sea-level rise (SLR). Worldwide, glaciers are expected to lose 26% (+1.5°C) to 41% (+4°C) of their mass by 2100, compared to 2015 (Rounce et al., 2023). In the Arctic, glacier mass loss enhances global warming through polar amplification (Manabe and Wetherald, 1975). A better understanding of Svalbard glacier change is therefore crucial to assess their future evolution and their impact on the climate system. The Global Climate Observing System (GCOS) has identified three essential climate variables (ECVs) for glaciers: glacier area, elevation change, and mass change; and four more for ice sheets and ice shelves: elevation and volume changes, velocity, and grounding line location and thickness. The aim of this study – calving front locations – is an important part of the glacier area ECV since Svalbard glaciers we study are constrained by terrain in all other directions.

## 1.2 Usage of early Landsat images

The MultiSpectral Scanner (MSS) instrument equipped Landsat 1–5, and was in use between 1972 and 1999 (and was briefly turned back on in 2012–2013), enabling retrieval of glacier area changes. For example, Williams (1987) successfully used two Landsat MSS images to measure the area of Iceland's largest ice cap (Vatnajökull) and to inventory 38 of its outlet glaciers. Yavaşlı et al. (2015) used Landsat MSS, along with other types of multispectral satellite data, to calculate the area change of glaciers in Turkey. Landsat MSS images have also been used in Svalbard for glacier studies. Dowdeswell (1984) used Landsat MSS images to map the ice cap margins of Nordaustlandet and compared them to maps from that time. He discovered that revisions of these maps were necessary, particularly on the East coast and at certain outlet glacier margins. He also mapped the snow line position and the basal melting through the observation of sediment plumes offshore. Hagen et al. (1993) finally, used Landsat 3 images, among other sources of data, to produce an inventory of Svalbard glaciers and report all known surge events in Svalbard. They documented the type, area, activity, and geometry of every Svalbard glacier larger than 1 km². This work was carried out as part of the World Glacier Inventory (WGI).

Automated glacier mapping techniques using band ratios (Hall et al., 1987; Rott, 1994; Bayr et al., 1994)) and the normalized difference snow index (NDSI; Hall et al., 1995) are not applicable to Landsat MSS images because of the lack of a SWIR band, which therefore need to be processed by hand. Deep-learning approaches allow overcoming this limitation (e.g. Li et al., 2024).

Li et al. (2024) used Landsat, Terra-ASTER and Sentinel-2A/B multispectral images, as well as Sentinel-1A/B images to train a deep learning model that produced a dataset of 124,919 glacier calving fronts for 149 marine-terminating glaciers in Svalbard between 1985 and 2023. Li et al. (2025) analyzed this dataset and observed seasonal cycles in front locations, with a several-month lag between peak retreat on Spitsbergen East and West coasts. Their observations are consistent with regional ocean warming patterns. After removing the seasonal variability, they calculate a retreating trend over the whole study time frame, with a rate of  $23.78 \, \mathrm{km}^2 \, \mathrm{yr}^{-1}$ . Li et al. (2024) and Li et al. (2025) provide a detailed historical monitoring of Svalbard glacier calving fronts. However, early Landsat images (Landsat 1–3) were not used in the production of the final dataset.

## 55 1.3 Usage of Declassified Intelligence Satellite Photographs (DISP)

Since 1995 and the declassification of 860,000 US military intelligence satellite photographs from the 1960s and 1970s, it has been possible to extend the time series of satellite remotely sensed glacier data. KH-4, 4A and 4B CORONA (cf. Sect. 2.2.1) photographs have been used for historical studies, especially in archaeology (Galiatsatos et al., 2004; Goossens et al., 2006) and more recently in glaciology (Ghuffar et al., 2022; Cooper et al., 2022; Racoviteanu et al., 2022). Cooper et al. (2022) used these images along with other sources of historical data for the study of surface mass balance (SMB) and frontal advance/retreat of outlet glaciers in the eastern margin of the Greenland ice sheet and their responses to air and ocean temperature changes. Racoviteanu et al. (2022) used orthocorrected CORONA stereo-images to produce digital elevation models (DEMs) of the Mansalu region of Nepal, and compared them to recent sources of data to study glacier SMB.

KH-5 ARGON (cf. Sect. 2.2.1) images have also been used for historical studies. Several techniques have been used to process these images: space-resection (Sohn and Dowman, 2000; Kim et al., 2001; Molnár et al., 2021), a sophisticated photogrammetric algorithm to retrieve the satellite position and attitude; and bundle-block adjustment (Wang et al., 2016; Ye et al., 2017), a technique that takes advantage of the overlap between two consecutive images to correct them simultaneously. Kim et al. (2001) used KH-5 ARGON images to map the ice shelf margins along Queen Maud Land, Antarctica, and compared them to more recent observations. Wang et al. (2016) used modern Worldview images to identify ground control points (GCPs) over the Larsen Ice Shelf (Antarctica) to create an orthocorrected version of ARGON images with bundle-block adjustment. They discovered that the acceleration that led to the collapse of Larsen B happened earlier than previously thought. Ye et al. (2017) proposed a new scheme to identify GCPs from modern image mosaic and DEM products, and achieved orthocorrection with a higher positioning accuracy than the ARGON image resolution. They used these images to produce a regional DEM of the Antarctic ice sheet. There is no published study using declassified intelligence satellite photographs (DISP) on Svalbard, despite the availability of several CORONA and ARGON images.

## 1.4 Usage of historical aerial images

Several aerial photograph mapping campaigns been carried out over Svalbard in the past century. Liestøl (1969) used historical photographs and written observations to produce an inventory of past glacier surges, and identified possible future surges. Geyman et al. (2022) took advantage of aerial images from 1936–1938 to generate a DEM of the archipelago, used to investigate historical mass balance changes of Svalbard glaciers in a 70-year time span. They discovered a strong correlation between the temperature and the mass loss rate: a  $1^{\circ}$ C increase in mean summer temperature corresponds to an area-normalized mass balance of  $-0.28\,\mathrm{m.yr^{-1}}$ . They also predict a doubling of mass loss by 2100, using a space-for-time substitution. Nuth et al. (2010) compared a variety of DEMs produced from different aerial campaigns to calculate Svalbard glacier elevation changes and contribution to SLR between the 1960s and the 2010s. They calculated a volume change of  $-0.36\pm0.02\,\mathrm{m\,w.e.\,yr^{-1}}$ , corresponding to a SLR of  $0.026\,\mathrm{mm\,yr^{-1}}$ .

The objective of this study is to bridge the gaps between historical studies of Svalbard glaciers, by retrieving glacier calving fronts from DISP (1962–1963) and Landsat (1976–1978) images, and comparing the observations to historical aerial image calving front data (1936–1938). In particular, this is the first study using DISP over Svalbard. It aims at comparing the front locations to assess their cumulative retreat or advance between the late 1930s and 1970s, which is a good indicator of glacier area change; and to document possible surge events that occurred during this period.

#### 2 Data and methods

#### 2.1 Landsat MSS images

Landsat is a scientific Earth-observation program that started in 1972 and is still ongoing. Instruments equipped on the Landsat satellites have evolved over time to provide observations in more spectral bands and with higher spatial resolution. In this study, we focus on the early stages of the Landsat program, when the MSS was the main instrument onboard. U.S. Geological Survey (2020a) provides an overview of the Landsat MSS data products. We identified sixteen usable images of Svalbard from 1976 and 1978. Each image covers a  $180 \text{ km} \times 180 \text{ km}$  area, and the ground-projected pixel size is 60 m. These images are freely available on the US Geological Survey (USGS) EarthExplorer website (https://earthexplorer.usgs.gov/) and can be downloaded in GeoTIFF format. Figure 1 gives an example of two of these images.

We produced false color images from bands 7, 5, and 4 using the *rioxarray* Python package. The original geolocation was off by multiple kilometers, which required correction. To do this, we selected about 20 GCPs for each image (cf. Table 1) in QGIS using the *Georeferencer* tool and applied a 2nd degree polynomial transformation.

#### 2.2 DISP

115

#### 2.2.1 DISP description

Between 1959 and 1984, the US intelligence agencies operated a series of reconnaissance satellites as part of a program called KeyHole (KH). These missions resulted in a series of 1,660,000 satellite images that were declassified in three stages in 1995, 2002, and 2011. These images are highly valuable for historical studies and are referred to as DISP in the remote sensing scientific community.

In this study, we focus on the images released in 1995, covering a period from 1960 to 1972. They were taken on panchromatic film, sent back to Earth in capsules that were retrieved mid-air by the US Air Force, and developed by the Central
Intelligence Agency. Today, a copy of these films is conserved by the USGS, which can provide high-quality scans for \$30 per
frame at the time of writing (U.S. Geological Survey, 2018). Once the films are scanned, the images remain freely available
on the EarthExplorer website. Several satellite systems were used to acquire these images, whose main characteristics are
summarized in Table 2 (Sohn and Dowman, 2000; U.S. Geological Survey, 2008).

For this study, we used eight KH-5 ARGON images, from missions 9034A (May 1962) and 9058A (August 1963). An example is given in Fig. 2. They were taken on 127 mm Kodak 3400 film, with a 76.2 mm focal length and an f/2.5 aperture

| Landsat MSS  |                | KH-5 ARGON   |                |  |
|--------------|----------------|--------------|----------------|--|
| Image        | Number of GCPs | Image        | Number of GCPs |  |
| 1976-07-12   | 15             | 1962-05-16-a | 178            |  |
| 1978-07-23   | 21             | 1962-05-16-b | 134            |  |
| 1976-07-16   | 16             | 1962-05-16-с | 267            |  |
| 1976-07-18-a | 24             | 1962-05-16-d | 241            |  |
| 1976-07-18-b | 24             | 1962-05-16-е | 252            |  |
| 1978-09-04   | 13             | 1963-08-29-a | 167            |  |
| 1978-09-22   | 21             | 1963-08-29-b | 164            |  |
| 1976-07-23   | 23             |              |                |  |
| 1976-07-09   | 26             |              |                |  |
| 1976-07-10   | 27             |              |                |  |
| 1976-07-17   | 21             |              |                |  |
| 1976-07-11   | 20             |              |                |  |
| 1976-06-06   | 22             |              |                |  |
| 1976-05-05   | 12             |              |                |  |
| 1978-07-25   | 22             |              |                |  |
| 1978-08-12   | 24             |              |                |  |

**Table 1.** Number of GCPs for Landsat MSS and KH-5 ARGON images georeferencing. For the sake of simplification, the images are identified by their date of acquisition. The full names are given in the appendix (Table A1).

| Mission designator | Satellite system | Area covered                   | <b>Ground resolution</b> | Comments              |
|--------------------|------------------|--------------------------------|--------------------------|-----------------------|
| KH-1, KH-2, KH-3,  | CORONA           | $200\times16~\mathrm{km}$      | 7.6 m                    | Single panoramic cam- |
| KH-4               |                  |                                |                          | era                   |
| KH-4A              | CORONA           | $200\times16~\mathrm{km}$      | 2.7 m                    | Two panoramic cameras |
| KH-4B              | CORONA           | $200\times16~\mathrm{km}$      | 1.8 m                    | Two panoramic cameras |
| KH-5               | ARGON            | $540 \times 540 \ \mathrm{km}$ | 140 m                    | Single mapping camera |
| KH-6               | LANYARD          | $14\times574~\mathrm{km}$      | 1.8 m                    | Single panoramic cam- |
|                    |                  |                                |                          | era                   |

 Table 2. Summary of the DISP satellite systems.

Figure 1. Example of Landsat MSS images (mosaic of two images taken in direct succession, left), and zoom on Tunabreen front (right). The full image names are given in appendix (Table A1). Background map: TopoSvalbard © Norwegian Polar Institute; image credit: U.S. Geological Survey

(Burnett, M. G., 1982). The satellite altitude was between about 300 and 600 km, and the scan resolution is  $7\mu m$  (U.S. Geological Survey, 2018; Molnár et al., 2021), resulting in a ground-projected pixel size of about 30 m on the digital copy. These images were taken in three different series (two on 16 May 1962, and one on 29 August 1963). Two consecutive images of the same series have an about 60% overlap.

## 2.2.2 KH-5 ARGON images processing

120

DISP are not georeferenced at all and present complex deformations that need to be corrected. For KH-5 ARGON images, these deformations are due to parallax, Earth's curvature, and lens distortion. Additionally, long-term storage on film introduces random deformations in the images. Previous studies focused on orthorectifying these images, and including all the necessary corrections in the process (cf. Sect. 1). We attempted to implement the corrections explained in Molnár et al. (2021), but the optimization process designed to retrieve the precise attitude parameters of the satellite failed. However, as we were only interested in retrieving glacier fronts that adjoin water surfaces, we estimated a full orthocorrection unnecessary. Instead, we georeferenced the images using QGIS. To take into account the more complex deformations of the KH-5 ARGON images, we used a 3rd degree polynomial transformation, with 100 to 200 GCPs per image. Table 1 (right) details the number of GCPs used for each image.

**Figure 2.** Example of KH-5 ARGON image (left) and zoom on Tunabreen front (right). Background map: TopoSvalbard © Norwegian Polar Institute; image credit: U.S. Geological Survey

## **2.3** 1936–1938 aerial photographs

In 1936 and 1938, two aerial expeditions were led by the Norwegian geologist Adolf Hoel over Svalbard. During these expeditions, 5,507 images were acquired, in order to produce topographic maps of Svalbard. These images have been used more recently to monitor elevation changes on Svalbard glaciers. Geyman et al. (2022) processed these images with a modern photogrammetrical workflow, and produced an orthorectified mosaic (Fig. 3) and a DEM. They also produced an updated 1936–1938 glacier outline inventory, which is used in this study together with the new 1960s and 1970s measurements.

## 2.4 Front digitization

Once we had georeferenced the Landsat and KH images, we digitized tidewater glacier fronts. Given the small number of usable images, we carried out this process manually in QGIS. Therefore, the accuracy of the front locations is, at best, the image resolution (60–140 m). On several Landsat images and on 1962 KH images, sea ice is present in the fjords, resulting in a lower accuracy. We retrieved the fronts from every image on which they were visible (Fig. 1 and 2, right panel). Therefore, for several glaciers, multiple mappings from the same year are available and can be used to assess the positioning uncertainty. In total, we made 467 Landsat and 526 KH observations, for 185 calving fronts on 171 glaciers.

Figure 3. 1936–1938 aerial images ortho-mosaic (left) and zoom on Tunabreen front (right). The yellow semitransparent areas represent the glacier outlines given by Geyman et al. (2022). Background map: TopoSvalbard © Norwegian Polar Institute; image credit: Norwegian Polar Institute

For the aerial photographs, Geyman et al. (2022) had already digitized glacier outlines. Therefore, we simplified the shapefile containing the glacier polygons to extract only the tidewater glacier fronts. 169 front observations are included in this dataset (Fig. 3, right panel). All Svalbard tidewater glaciers are covered, except the easternmost part of Nordaustlandet.

We then enriched the datasets with the images' names, dates, missions, RGI indexes version 6 and 7, and flow directions to facilitate data analysis. Since some glaciers in the dataset have several calving fronts (e.g. Rijpbreen), we created an index to precisely identify each front. In some places, several glaciers merge to form a unique front. In these cases, we split front observations so that one portion of the front is associated with each glacier, following the RGI and 1936–1938 partition. An example is given in Hornsund, Fig. 4. We used the regular WGS84 ellipsoid projection for the Landsat and KH fronts, while we kept the original CRS (EPSG:32633, WGS84 - UTM zone 33N) for the aerial image fronts.

# 2.5 Glacier frontal changes characterization

155

This study focuses on quantifying front advances and retreats, and therefore at simplifying front geometry changes into single numbers, to quantify their motions along the glacier flows. We used the extrapolated centerline method, introduced by Lea et al. (2014), as an improvement of the centerline method. In the regular centerline method, the distance between two front observations is simply calculated as the distance between the intersection of the fronts and the glacier centerline, resulting in

165

**Figure 4.** Example of split Landsat front in Hornsund. RGI (plain) and 1936 (semi-transparent) outlines are also displayed to show the boundaries between the different glaciers. Background map: TopoSvalbard © Norwegian Polar Institute

a loss of all the information about the front geometry. The extrapolated centerline method uses inverse distance weighting to define the position of every front point on the glacier centerline. The final position of the front is then defined as the mean value of the positions of its points (cf. Lea et al., 2014, for implementation details; the method of this study is identical to theirs). We digitized centerlines by hand, due to the lack of concomitant topography and flow information.

We compared the results obtained through this method to those calculated with the box method (Lea et al., 2014) when possible. We found a strong agreement between both methods. Since the extrapolated centerline method is designed to improve tracking of glaciers with changes in front width and non-rectilinear flows (Lea et al., 2014), we chose it to calculate all the results.

We retrieved area change by multiplying glacier front displacements (calculated with the previously presented methods) and the average lengths of the fronts (Li et al., 2025). In this study, we calculated cumulative area change by adding the contributions of all glaciers together.

#### 2.6 Uncertainties

For the fronts that were retrieved from KH-5 ARGON and Landsat MSS images, we assessed the accuracy of the front locations by comparing fronts that were digitized from images dating from the same year. We consider this method to be reliable as Landsat MSS images were all taken during summer, and KH-5 ARGON images were taken in series on the same day. Therefore, the true location of a front should be the same on two photographs from the same year. Harcourt et al. (2025) identified three marine-terminating glaciers that were possibly in active surge phase when the Landsat images were taken:

Allfarvegen (1970–1980), Rijpbreen (1938–1992) and Bodleybreen (1976–1980). We therefore excluded these glaciers from the Landsat uncertainty calculations.

We separated the uncertainties into three types: on front positions, on front displacements and on cumulative area change. Since the qualities of the georeferencing and the front delineation depend mainly on the image quality, we calculated one value for each uncertainty type per year or possible combination of years (cf. Appendix for calculation details).

#### 180 3 Results

185

## 3.1 Average frontal change

Figure 5a displays time series of front positions (corrected to set their values to zero in 1936–1938). The mean glacier front position is also shown, as well as its associated uncertainties. Svalbard tidewater glaciers retreated by  $26.3\,\mathrm{m\,yr^{-1}}$  ( $R^2=0.89$ ) between 1936 and 1978. This length loss represents  $1140\pm136\,\mathrm{m}$  in 1978, compared to 1936–1938. The graphs show a faster retreat rate between 1936–1938 and 1962–1963, followed by a plateau until 1976–1978. The retreat rate seems to increase again between 1976 and 1978. However, the lack of data points between 1936–1938 and 1962–1963, and the overall high uncertainties (especially in 1962–1963), make it difficult to draw a robust conclusion. One can also see the diversity of behaviors of Svalbard tidewater glaciers (gray lines on Fig. 5a): most are retreating, but some are advancing, and they have a wide variety of retreat/advance rates.

Figure 5b shows the cumulative area change due to calving front displacements for all measured glaciers between 1936 and 1978. In 1978, the area loss is  $642 \pm 167 \,\mathrm{km}^2$ , with an average area loss rate of  $16.0 \,\mathrm{km}^2 \,\mathrm{yr}^{-1}$  ( $R^2 = 0.94$ ).

#### 3.2 Individual glacier events

To identify significant retreat and advance events, we compared all front observations together, for each glacier. Then, we identified pairs of fronts whose distances were higher than the recombined uncertainties presented in Appendix B2. These events correspond to front advances or retreats with at least a 95% confidence. Figure 6 shows the distance distributions and

Figure 5. Time series of front locations (a) and cumulative area loss (b). The locations have been corrected so that the origin corresponds to 1936–1938. The gray lines represent individual glacier fronts, and the red line corresponds to the mean glacier front position. The error bars define the 95% confidence interval ( $2\sigma$ ) based on the corrected standard deviations calculated in Appendix B1 (a) and Appendix B3 (b).

highlights the significant events. Table C1 in the appendix summarizes the number of observations used for each distribution, and shows their mean values and standard deviations.

Several observations can be made about these distributions. First, the mean values are all negative, meaning that on average, the fronts are observed to retreat. Second, dispersions are higher when the time span between front observations is greater. Finally, more significant observations are made when the uncertainties on front locations for both years are low, and when the time span between the observations is high.

We identify a total of 15 advances and 122 retreats. A 16<sup>th</sup> advance of Storisstraumen was detected between 1976 and 1978, but this is likely due to the georeferencing uncertainty of the Landsat images. The uncertainty is high in this region due to the lack of stable terrain to be used as georeference points. Therefore, this glacier has been removed from the results.

Figure 6. Distribution of distances between two fronts for every possible year combination. Colors represent the degree of confidence with which a retreat or advance of the front was detected: light green  $-2\sigma$  (95%); dark green  $-3\sigma$  (99%). Uncertainties are based on the recombined variances calculated in Appendix B2.

On Fig. 7, maps for all filtered advance events are displayed. One can observe on these maps that a specific front can show both behaviors (e.g. Hilopenbreen retreated between 1938 and 1962–1963 and then advanced between 1962–1963 and 1976–1978, or Freemanbreen, which advanced between 1936 and 1963 and then retreated between 1963 and 1976). Also, an advance/retreat might not be uniform along the front (e.g. Stonebreen), which we bring up further in the discussion.

Because of the high uncertainties (especially in 1962–1963), it is necessary to calculate the distance between fronts for every possible pair of years, and not only between consecutive observations. For example, the advance of Recherchebreen (Fig. 7) was detected only between 1936 and 1978, although it appears that the advance happened between 1936 and 1962, and then the glacier retreated between 1962 and 1978.

## 4 Discussion

210

220

## 4.1 Average frontal change

We calculated an average retreat rate of  $26.3 \,\mathrm{m\,yr^{-1}}$  ( $R^2 = 0.89$ ) for the 171 tracked glaciers between 1936 and 1978. Li et al. (2025) have found a wide variety of front change rates for 1985–2023, although the most observed is  $25-50 \,\mathrm{m\,yr^{-1}}$ , which is of the same order of magnitude as the one we calculated for a prior time frame.

We found a cumulative area loss rate of  $16.0\,\mathrm{km^2\,yr^{-1}}$  ( $R^2=0.94$ ) for all the tracked glaciers between 1936 and 1978, which is consistent, though lower, compared to the value of  $23.78\,\mathrm{km^2\,yr^{-1}}$  calculated by Li et al. (2025) for the 1985–2023 period, and the value of  $26.79\pm0.54\,\mathrm{km^2\,yr^{-1}}$  calculated by Kochtitzky and Copland (2022) for 2000–2020. This could

**Figure 7.** Maps of the main front advance events. Fronts from aerial images (1936–1938) are in green, KH-5 ARGON fronts (1962–1963) are in red and Landsat MSS fronts (1976–1978) are in blue.

| Glacier name                | Date of detected advance | Surge documentation | Surge period |
|-----------------------------|--------------------------|---------------------|--------------|
| Emmabreen                   | 1936–1963                | No documentation    | no date      |
| Nansenbreen                 | 1936–1963                | LI, HA, BL          | 1946–1948    |
| Tunabreen                   | 1963–1976                | HA                  | 1970         |
| Recherchebreen              | 1936–1978                | HA, BL              | 1945         |
| Anna Margrethebreen         | 1963–1976                | HA, BL              | 1970         |
| Emil'janovbreen & Spælbreen | 1963–1976                | LE, HA, BL          | 1970–1971    |
| Allfarvegen                 | 1976–1978                | BL                  | 1970–1980    |
| Hinlopenbreen & Oslobreen   | 1963–1976                | LE, HA, BL          | 1969–1971    |
| Kosterbreen                 | 1938–1976                | LE                  | 1956–1970    |
| Freemanbreen                | 1936–1963                | LE, HA, BL          | 1955–1956    |
| Stonebreen                  | 1936–1963                | LE, HA, BL          | 1936–1971    |
| Schweigaardbreen            | 1938–1976                | No documentation    | no date      |
| Fonndalsbreen               | 1938–1976                | No documentation    | no date      |
| Rijpbreen                   | 1938–1976                | HA, BL              | 1938         |
| Bodleybreen                 | 1938–1976                | LE, HA, BL          | 1973         |

**Table 3.** Advance events detected, and corresponding documented glacier surges. The letters correspond to the references: Liestøl (1969) (LI), Lefauconnier and Hagen (1991) (LE), Hagen et al. (1993) (HA) and Błaszczyk et al. (2009) (BL). Glaciers in bold indicate the events for which additional information has been provided in this study.

indicate an acceleration of the area loss of Svalbard marine-terminating glaciers due to calving front retreats over the past century. However, several differences in the methods to calculate these numbers remain: like Li et al. (2025), we calculate the change in area as the product of the retreat length and the glacier width, but we keep surge-type glaciers in the dataset for this calculation, and do not take into account the seasonal variability. The number given by Kochtitzky and Copland (2022) includes Svalbard and Jan Mayen, and surge-type glaciers are kept in this dataset. However, the outline of the area that changed from glacier-covered to not glacier-covered was manually digitized, instead of being calculated as a product of retreats' lengths and glacier's widths.

#### 4.2 Individual glacier events

230

We detected a total of 13 glacier surges and 2 other advances (cf. Sect 4.2.2). Table 3 lists all the surge/advance events we detected, and reports the studies that mention them, as well as the time windows they identified. For two glaciers (Allfarvegen and Stonebreen), we narrowed the time window in which the surge was known to happen, and we discovered one new undocumented surge of Emmabreen.

### 4.2.1 Surge of Emmabreen between 1936 and 1963

The Emmabreen surge appears clearly on Fig. 7 & 8: in 1963, the glacier front reaches the fjord mouth, whereas in the 1936 aerial and 1976 Landsat images, the front lies further into the fjord. The front advanced by 450 m between 1936 and 1968.

Two fronts from 1963 are available in the dataset, about 500 m away from each other. However, the front that was digitized from the 1963-09-29-b image was on the edge of the photograph, where the deformations (due to Earth's curvature and lens distortion) are stronger and the georeferencing less accurate. We did not take this front into account in the calculations as it does not intersect the centerline for this glacier, and therefore did not display it on Fig. 7.

240 RGI 7.0 Consortium (2023) did not categorize Emmabreen as a surge-type glacier. Nevertheless, Harcourt et al. (2025) reported a surge between 2011 and 2021. Therefore, we conclude that Emmabreen has surged twice in the last century; between 1936 and 1963, and in the 2010s.

**Figure 8.** 1936–1938 aerial photograph (left), 1963 DISP (center) and 1976 Landsat MSS (right) images for the Emmabreen surge. The green line (resp. red, blue) corresponds to the 1936-1938 (resp. 1963, 1976) front. Image credits: Norwegian Polar Institute (aerial image) and U.S. Geological Survey (DISP and Landsat).

## 4.2.2 Advances of Schweigaardbreen and Fonndalsbreen

Schweigaardbreen is not documented as a surge-type glacier, but the presence of crevasse-squeeze ridge networks seen in recent aerial images (Lovell and Boston, 2017) indicates a possible misclassification. The images shown in Fig. 9a show an advance of about 400 m between 1938 and 1976. The 1962 DISP images are too inaccurate to narrow down the time frame of the advance. Therefore, we conclude that our observations are compatible with a possible surge of Schweigaardbreen. Similarly, we registered a 541 m advance of the neighboring Fonndalsbreen between the same years (Fig. 9b). They share a vague resemblance to recent reports of multi-decadal advances of glaciers in Arctic Canada (Medrzycka et al., 2019; Lauzon et al., 2024). However, since the advances of both these glaciers coincide with recent regional mass gains (Moholdt et al., 2010)

and there is a tendency of glaciers in this region to extend further today than in 1938 (Geyman et al., 2022), we refrain from calling these events surges and instead only report on their advances.

**Figure 9.** 1938 aerial photograph (left) and 1976 Landsat MSS (right) images for the Schweigaardbreen (a) and Fonndalsbreen (b) advances. The green lines (resp. blue) correspond to the 1938 (resp. 1976) fronts. Note the different scales for the two glaciers. Image credits: Norwegian Polar Institute (aerial images) and U.S. Geological Survey (Landsat).

## 4.2.3 Narrowing the timeframes of the surges of Stonebreen and Allfarvegen

An advance of 1664 m of the northern tongue of Stonebreen between 1936 and 1963 can also be observed in Fig. 7 & 10. This surge was already documented by Lefauconnier and Hagen (1991), Hagen et al. (1993) and Błaszczyk et al. (2009), and its known time frame was 1936–1971. Our observations narrow down this time frame to 1936–1963.

A complicating factor for this event when comparing it with a modern surge that just recently terminated (Mannerfelt et al., 2025) is that the two surges occurred in two different parts of the ice body. This is important for studies focused on "return

times" of glacier surges; the Stonebreen surge has not necessarily reoccurred but instead started in a new unique location. In other words, our study shows that the two surges of Stonebreen are only related through naming convention of the ice cap.

**Figure 10.** 1936 aerial photograph (left), DISP (center) and Landsat MSS (right) images for the Stonebreen surge. The green line (resp. red, blue) corresponds to the 1936 (resp. 1963, 1976) front. Image credits: Norwegian Polar Institute (aerial image) and U.S. Geological Survey (DISP and Landsat).

Finally, we observe an advance of 380 m of Allfarvegen between 1976 and 1978. A surge of this glacier in the 1970s was documented by Błaszczyk et al. (2009). Our observations narrow down this time frame to 1976–1978.

**Figure 11.** 1938 aerial photograph (left), 1976 (center) and 1978 Landsat MSS (right) images for the Allfarvegen surge. The green line (resp. blue, dashed blue) corresponds to the 1938 (resp. 1976, 1978) front. Image credits: Norwegian Polar Institute (aerial image) and U.S. Geological Survey (DISP and Landsat).

## 4.2.4 Retreats of Negribreen and Nathorstbreen 1938–1978

Among the 122 significant retreats detected, Negribreen's and Nathorstbreen's ones are the two most significant, with values of about 7300 m and 7600 m, respectively. Figure 12 shows the satellite and aerial images corresponding to these events. An important surge of Negribreen is known during 1935–1936 (Lefauconnier and Hagen, 1991; Hagen et al., 1993). Nathorstbreen surged in 1890, as well as Liestølbreen and Doktorbreen – two neighboring glaciers whose fronts merge with Nathorstbreen's one – surged between 1870 and 1880 (Harcourt et al., 2025).

**Figure 12.** 1936 aerial photograph (left), 1962–1963 DISP (center) and 1976–1978 Landsat MSS (right) images for the Negribreen (a) and Nathrostbreen (b) retreats. The green lines (resp. red, blue) correspond to the 1936 (resp. 1962–1963, 1976–1978) fronts. Note the different scales for different glaciers. Image credits: Norwegian Polar Institute (aerial images) and U.S. Geological Survey (DISP and Landsat).

### 4.3 Use of archived satellite images

In this study, we used archived satellite imagery to determine glacier calving front locations on Svalbard. However, it is likely that the dataset also contains geophysical information that is outside the scope of our study. A natural example of such information is the extent of ground-terminating glaciers. It is possible that the images show previously undocumented surges of ground terminating glaciers, or at least can be used to better constrain the times of past surges in the same manner as we have done in our study for marine terminating ones. To facilitate future research, we have made all the data available, including the georeferenced satellite images. Furthermore, this study has shown that the KH-5 ARGON images are useful for the study of the Svalbard glaciers. Thus, we suggest that their use for other remote geographic areas be explored.

#### 5 Conclusions

In this study, we have used three different types of aerial and satellite images to investigate historical changes in Svalbard tidewater glacier fronts between 1936 and 1978. We have successfully retrieved calving fronts from Landsat MSS multispectral images and DISP, and compared our observations to the glacier outlines from 1936–1938 to assess the retreat or advance of the glaciers. To the best of our knowledge, this is the first study to use DISP over Svalbard.

An analysis of the average glacier behavior over the whole study period yields an average retreat rate of  $26.3\,\mathrm{m\,yr^{-1}}$  ( $R^2=0.89$ ), which agrees with the value of  $25\text{-}50\,\mathrm{m\,yr^{-1}}$  found by Li et al. (2025) for a subsequent time period. We calculated a cumulative area loss rate of  $16.0\,\mathrm{km^2\,yr^{-1}}$  ( $R^2=0.94$ ). This loss is likely due to increasing air and ocean temperatures in Svalbard, and strongly agrees with the value of  $23.78\,\mathrm{km^2\,yr^{-1}}$  calculated by Li et al. (2025) for 1985--2023, and the value of  $26.79\pm0.54\,\mathrm{km^2\,yr^{-1}}$  found by Kochtitzky and Copland (2022) for 2000--2020.

Finally, we conducted an extensive comparison of all the fronts and identified 13 glacier surges. For four of these glaciers, our study brings new information: we discovered an undocumented surge of Emmabreen (1936–1963) and narrowed down the time frames of two known surges: Allfarvegen (1976–1978) and Stonebreen (1936–1963). We also documented an advance of Schweigaardbreen and Fonndalsbreen between 1938 and 1976.

To continue this study, future work should focus on improving the precision of the observations, especially for those retrieved from DISP. This can be done by orthorectifying the images, but may require access to calibration data. The analysis could be extended as well, by assessing the spatial variability of the average retreat rate. The dataset containing all the calving fronts is freely available.

295 Code and data availability. All the data used in this study is available on Zenodo (https://zenodo.org/records/17391880). This dataset includes the Landsat MSS and KH-5 ARGON satellite images (TIF files), the GCPs used to georeference them (POINTS files), all the calving fronts (shapefiles, including 1936-1938 fronts adapted from Geyman et al. (2022)), and all the boxes and centerlines used to calculate distances.

The code used for the data analysis if fully available on Github (https://github.com/lorisdanjou/Svalbard\_calving\_fronts\_1936-1978\_data\_analysis).

# 300 Appendix A: Full names of the satellite images

To facilitate the reading of this paper, we have shortened the names of the satellite images. Table A1 (resp. Table A2) provides the full names of all the used Landsat MSS (resp. KH-5 ARGON) images. These names are the ones used in the EarthExplorer platform.

| Full image name                          | Short image name |
|------------------------------------------|------------------|
| LM02_L1GS_224004_19760712_20200907_02_T2 | 1976-07-12       |
| LM02_L1GS_227005_19780723_20210610_02_T2 | 1978-07-23       |
| LM02_L1GS_228003_19760716_20200907_02_T2 | 1976-07-16       |
| LM02_L1GS_230003_19760718_20200907_02_T2 | 1976-07-18-a     |
| LM02_L1GS_230004_19760718_20200907_02_T2 | 1976-07-18-b     |
| LM02_L1GS_234002_19780904_20210610_02_T2 | 1978-09-04       |
| LM02_L1GS_234004_19780922_20210610_02_T2 | 1978-09-22       |
| LM02_L1GS_235002_19760723_20200907_02_T2 | 1976-07-23       |
| LM02_L1GS_239002_19760709_20200907_02_T2 | 1976-07-09       |
| LM02_L1GS_240002_19760710_20200907_02_T2 | 1976-07-10       |
| LM02_L1GS_241001_19760717_20200907_02_T2 | 1976-07-17       |
| LM02_L1GS_241002_19760711_20200907_02_T2 | 1976-07-11       |
| LM02_L1GS_242002_19760606_20200907_02_T2 | 1976-06-06       |
| LM02_L1GS_245002_19760505_20200907_02_T2 | 1976-05-05       |
| LM03_L1GS_238002_19780725_20210611_02_T2 | 1978-07-25       |
| LM03_L1GS_238003_19780812_20210611_02_T2 | 1978-08-12       |

Table A1. Full names of the Landsat MSS images used in this study (U.S. Geological Survey, 2020b).

| Full image name  | Short image name |
|------------------|------------------|
| DS09034A007MC018 | 1962-05-16-a     |
| DS09034A007MC019 | 1962-05-16-b     |
| DS09034A008MC020 | 1962-05-16-c     |
| DS09034A008MC021 | 1962-05-16-d     |
| DS09034A008MC022 | 1962-05-16-e     |
| DS09058A024MC012 | 1963-08-29-a     |
| DS09058A024MC013 | 1963-08-29-b     |

Table A2. Full names of the KH-5 ARGON images used in this study (U.S. Geological Survey, 2021).

Figure B1. Distance histograms between fronts retrieved from KH-5 ARGON (red) and Landsat MSS (blue), ordered by year. The distances were calculated with the extrapolated centerline method. The standard deviations ( $\sigma'$ ), as well as the corrected standard deviations on front positions ( $\sigma = \frac{\sigma'}{\sqrt{2}}$ ) are indicated in the top-left corner of each plot.

## **Appendix B: Details of uncertainty calculations**

# **B1** Front positions

305

310

Figure B1 displays the distributions of distances calculated between fronts dating from the same year.

For KH-5 ARGON images, one can observe that more observations were available for the year 1962 (five images in 1962, only two in 1963). One can also see that distances are more dispersed in 1962. This is maybe due to the greater number of available observations, but more certainly to the sea ice conditions at the time of capture: in May 1962, a lot of sea ice was present in the fjords, and therefore glacier fronts were more difficult to distinguish. For Landsat MSS images, more observations were available in 1976. Due to the overall higher quality of the images compared to KH-5 ARGON, the distribution is less dispersed. Sea ice conditions were similar on pictures from both years, and false colors made glaciers easier to distinguish from the background.

These series seem to follow centered normal distributions. The standard deviations (noted  $\sigma'$ ) are indicated on Fig. B1 for each distribution. These standard deviations are calculated for distances between two fronts, which equally contribute to the uncertainty. Therefore, they must be divided by  $\sqrt{2}$  to obtain the standard deviations on front locations. The corrected standard deviations (noted  $\sigma$ ) are indicated as well on Fig. B1.

For the 1936-1938 fronts, the uncertainty on glacier outlines was already assessed by Geyman et al. (2022), and its value is 30 m.

#### 320 B2 Front displacements

The standard deviations on front positions can be recombined to assess the uncertainty on the distance calculated between the measures of two different years. Indeed, if  $\sigma_{\text{year 1}}$  and  $\sigma_{\text{year 2}}$  are the position uncertainties for two different years, then the uncertainty on front displacement  $\sigma_{\text{year 1, year 2}}$  between these two years is:

$$\sigma_{\text{year 1, year 2}} = \sqrt{\sigma_{\text{year 1}}^2 + \sigma_{\text{year 2}}^2}$$

325

All the possible standard deviations are summarized in Table B1.

| Year 1    | Year 2 | $\sigma_{ m year1,year2}$ |
|-----------|--------|---------------------------|
| 1936-1938 | 1962   | 360 m                     |
| 1936-1938 | 1963   | 182 m                     |
| 1936-1938 | 1976   | 134 m                     |
| 1936-1938 | 1978   | 71 m                      |
| 1962      | 1963   | 402 m                     |
| 1962      | 1976   | 383 m                     |
| 1962      | 1978   | 366 m                     |
| 1963      | 1976   | 224 m                     |
| 1963      | 1978   | 193 m                     |
| 1976      | 1978   | 149 m                     |

Table B1. Standard deviations for front distances, measured with extrapolated centerline method.

## **B3** Area change

Uncertainties on cumulative area change can be calculated using the uncertainties on front positions ( $\sigma$ , Fig. B1) and the standard deviation on front widths.

Let  $A_{ij}$  be a random variable representing the area change for front i between years j and j+1. By definition,  $A_{ij} = W_{ij}D_{ij}$ , where  $W_{ij}$  is the width of front i for year j, and  $D_{ij}$  is the displacement of the front i between years j ad j+1.

Thus, the cumulative area loss for all glaciers at year k is:

$$A_k = \sum_{j=1}^k \sum_i A_{ij}$$
$$= \sum_{j=1}^k \sum_i W_{ij} D_{ij}.$$

Therefore,

335 
$$\varepsilon_k^2 := \operatorname{Var}(A_k) = \operatorname{Var}(\sum_{j=1}^k \sum_i A_{ij})$$

$$= \sum_{j=1}^k \sum_i \operatorname{Var}(A_{ij})$$

$$= \sum_{j=1}^k \sum_i \operatorname{Var}(W_{ij}D_{ij})$$

$$= \sum_{j=1}^k \sum_i \left(\operatorname{Var}(D_{ij})\mathbb{E}(W_{ij}^2) + \operatorname{Var}(W_{ij})\mathbb{E}(D_{ij})^2\right)$$

$$= \sum_{i=1}^k \sum_i \left(\sigma_{j,j+1}^2 \mathbb{E}(W_{ij}^2) + \operatorname{Var}(W_{ij})\mathbb{E}(D_{ij})^2\right)$$

Where:

340

- $\varepsilon_k$  is the standard deviation on cumulative area change for year k,
- $\sigma_{j,j+1}$  is the standard deviation on front displacements between year j and years j+1,
- $\mathbb{E}(W_{ij}^2)$  is average squared width of front i for year j,
- $Var(W_{ij})$  is the variance of front i width for year j, and
  - $\mathbb{E}(D_{ij})$  is the average front i displacement between years j and j+1.

This calculation is equivalent to a sum of relative uncertainties. The error bars on Fig. 5 correspond to the 95% confidence interval ( $2\varepsilon_k$ ).

## Appendix C: Details on distances distributions

Table C1 provides more details on the distances distributions shown on Fig. 6.

| Year 1 | Year 2 | Number of observa- | Mean    | Standard deviation |
|--------|--------|--------------------|---------|--------------------|
|        |        | tions              |         |                    |
| 1962   | 1962   | 115                | -915 m  | 1589 m             |
| 1936   | 1963   | 82                 | -917 m  | 1625 m             |
| 1936   | 1976   | 143                | -994 m  | 1729 m             |
| 1936   | 1978   | 100                | -1086 m | 1857 m             |
| 1962   | 1963   | 60                 | -48 m   | 279 m              |
| 1962   | 1976   | 107                | -227 m  | 762 m              |
| 1962   | 1978   | 80                 | -231 m  | 903 m              |
| 1963   | 1976   | 72                 | -156 m  | 838 m              |
| 1963   | 1978   | 54                 | -121 m  | 949 m              |
| 1976   | 1978   | 93                 | -57 m   | 180 m              |

Table C1. Number of observations, mean value, and standard deviation for each distribution shown in Fig.6.

Author contributions. This paper is based on the Master's thesis project of LD which he carried out under the supervision of ER. LD processed the satellite data, digitized the glacier fronts and analysed the dataset. ER and ESM provided support and supervision. All three contributed to the writing of the manuscript.

Competing interests. The authors declare that they have no conflict of interest.

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
