# Peer review of "Svalbard glacier calving front retrievals during the 1960s and 1970s from archived Landsat and declassified intelligence satellite photographs"

_EGUsphere, 2025_

## Referee Comment (RC1)

Review of 'Svalbard glacier calving front retrievals during the 1960s and 1970s from archived Landsat and declassified intelligence satellite photographs'

The manuscript titled 'Svalbard glacier calving front retrievals during the 1960s and 1970s from archived Landsat and declassified intelligence satellite photographs' presents a valuable reconstruction of calving front changes for all Svalbard tidewater glaciers using historical satellite data (KH-5 ARGON and Landsat) and aerial imagery from the 1930s. This is the first study using Declassified Intelligence Satellite Photographs (DISP) over Svalbard, and the authors conduct a thorough and detailed analysis of the accuracy of remote sensing source materials. Despite the limitations of the DISP data, the results provide valuable insights into the average retreat for the entire archipelago between the 1930s, 1960s and late 1970s. Furthermore, it provides detailed information on the behavior of individual glaciers. The freely available dataset of georeferenced imagery and digitized calving fronts represents a significant contribution that can be highly valuable for future glaciological and climate-related studies.

Based on the review, it is recommended that the manuscript be considered for acceptance with minor revisions. I found it interesting that the retreat in the study period is lower than the values of other authors for slightly later periods. As widely recognized, sea temperature, air temperatures and sea ice cover are among the most significant factors influencing glacier front positions. I suggest underlining it in the paper and discussing the results with the climatic data and reanalysis for the region.

In my opinion, all three glaciers, Emmabreen, Schweigaardbreen, and Fonndalsbreen, share similar characteristics; however, the presented results and discussion are insufficient to determine whether they are surging glaciers. I would suggest classifying all of them as advancing glaciers with the possibility of surging.

Additional comments include maintaining a consistent chronology throughout the text and in the legends of figures, as well as avoiding short paragraphs (two to three sentences) and combining them effectively to present your results clearly. Please also consider including the aerial photos from the 1930s in the manuscript title, as they are a primary source in the presented research.

Minor comments:

L2 and the title - please consider changing the word 'retrieve' to 'obtain/digitalize' (or another alternative) as an option.

L3 – The paper does not discuss geophysical analyses of the calving fronts. Please rewrite this sentence

L25 – Explain what (+1.5°C) and (+4°C) mean.

L25-27 – The statements are partially inaccurate simplifications and may be interpreted as implying that glacier mass loss acts as a driver of global warming. However, glacier mass loss primarily represents a response to climate warming and may subsequently contribute to local and regional amplification of warming through positive feedbacks, mainly related to changes in surface albedo. Please consider rewriting.

L29 – Avoid using acronyms (e.g., ECV) when they are used in the text only once or twice. ECV is not necessary in L 31.

L32 – '...since Svalbard glaciers we study are constrained by terrain in all other directions.' Actually, glaciers also change their area in the upper parts, which is especially visible in long-term analyses. Please consider rewriting or omitting that part.

L33 – I would suggest describing data sources chronologically, starting from aerial photos

L52 – What do you mean by 'a retreating trend' here?

L69 – Worldview => WorldView

L77 – campaigns been => campaigns have been

L82 – Geyman et al. (2022) give units in water equivalent (−0.28 m yr$^{-1}$ of water equivalent)

L86 – The word "historical" may suggest that the research itself is old, whereas the studies are sometimes modern but use older data. A clearer phrasing might be "..to fill gaps in past observations of Svalbard glaciers..."

L88 – Regarding years 1936-38, please mention here the orthophotos of Geyman et al. (2022)

L 104 – Please consider rewriting Section 2.2.1 slightly. Actually, you do not use images covering the period from 1960 to 1972 in your studies (L 109), but only ARGON data from 1962 and 1963. Leave in this section only information relevant to the ARGON mission. It is not clear from the table and text what the final resolution of pixels is: 30 m or 140 m? Is Table 2 necessary? It suggests that you use all this data in your research. Consider reorganizing chapters 2.1 - 2.3 to describe data in chronological order.

Figure 1 – Consider rewriting: 'a mosaic generated from two consecutively acquired images'. Here, and in other figures, please add a dot at the end of each sentence.

L123 – Consider changing 'deformations are due to parallax' to 'deformations arise from terrain-induced parallax...' to underline that the mountainous terrain of Svalbard influences deformation of images.

L126-129 – It is not clear how you processed the images ('...a full orthocorrection unnecessary. Instead...'). Was orthocorrection or only a 3rd-degree polynomial transformation applied?

L148 – If you cite any name, it should be shown on the map or figure. Please show Rijpbreen in the image as an example or remove the name from the text.

L154-155 – The whole sentence is not clear. What is the main aim? what is simplified? do you really quantify the glacier motion or just front position changes?

Figure 4. Unfamiliar readers do not know where Hornsund is. Please present it on the Svalbard map, e.g., in Figure 1. Please rewrite the sentence 'Example of split Landsat front in Hornsund' to be more understandable, e.g., 'Example of split of glacier fronts in Hornsund (background: Landsat)'. Add the date of glacier fronts from RGI. Reorder the legend to be chronological. Cite the RGI in the figure caption. Change Mendeleevbreen to Mendelejevbreen.

L182 – Is the 'y' axis correct in Figure 5? (in L265 you mention retreat 7300 m and 7600 m)

L184 – What do you mean by 'This length loss'? Is the sum of the linear retreat of all glaciers?

L185 and further – Do you analyze the retreat rate? or just faster/slower retreat?

L191 – Maybe start the sentence with 'The area loss between 1936-1978 was...'

L201 – 'time span ...is high' – you mean short? long?

L202 – add the studied time span in the first sentence.

L206 – What do you mean by 'both"? Rewrite the sentence to explain the two behaviors in the main part of the sentence, not in the brackets.

The entire section 3.2 – please avoid using short paragraphs and try to combine them properly to present your results effectively.

L210-212 – This sentence is not logical. Why was the advance of Recherchebreen detected only between 1936 and 1978? in Figure 7 it looks like advance was between 1936-1962?

Figure 7. Please reorder the legend chronologically and present the positions of all glaciers on the Svalbard map.

Section 4.1 – Cite the accuracy of your results. It is worth noting that your findings are slightly lower than those of Li et al. (2025) and Kochtitzky and Copland (2022). This is a very interesting finding, as your research covers an earlier period. Please discuss it in conjunction with the climate in the period of your study and that of the other authors.

L225 – Be precise in the last sentence. Do you describe your or Kochitzky's methods?

Section 4.2 – I miss the consistency in this discussion and conclusions. On what basis were the three glaciers (Emmabreen, Schweigaardbreen and Fonndalsbreen) divided into two separate categories, surging and advancing? In my opinion, you classify Emmabreen as a surging glacier just because it was mentioned as surging later in time (see L241) by Harcourt et al. (2025). But Lovell and Boston (2017) also suggest that Schweigaardbreen would be a surge-type. I agree that the positive mass balance would obscure the interpretation of front position changes for Schweigaardbreen and Fonndalsbreen, but the study period is very long, making it difficult to prove this. It is also worth noting in the manuscript that both glaciers originate from the ice cap Austfonna, which can influence glacier dynamics. Anyway, to document glacier surging, further studies are needed (DEM differentiation, velocity measurements or other indirect evidence). As you do not present any such evidence in all three cases, I would suggest combining sections 4.2.1 and 4.2.2 into one: 'Advances of Emmabreen, Schweigaardbreen and Fonndalsbreen', with the note that a surge of all three glaciers may have occurred during the period under study.

L232 – for consistency with the first sentence in the paragraph, shouldn't be 'cf. Sect. 4.2.1' cited here as well?

L234, 235 – I suggest using the term 'bay' instead of 'fjord' in this case

L237-238 – cf. L123

L246 – You write: ' The 1962 DISP images are too inaccurate to narrow down the time frame of the advance.' So why do you present the 1962/63 position of the glacier in Fig. 7? It suggests that the surge was before 1962/63.

L247 – 'Therefore, we conclude that...' – the sentence is not logically connected with the previous sentence.

L248 – 'They share a vague resemblance..' – the sentence is not logically connected with the previous sentence. What do you mean by 'they'? Comparison with the Canadian Arctic glacier is not relevant here, in my opinion.

L282-286 – Clarify 'subsequent time period'. Do not repeat all numbers from results and discussion, but try to describe the main outcomes - see comments to Section 4.1.

L312 – false color => false color composite